# Ultrasonic-Vibration-Assisted Waterjet Drilling of [0/45/−45/90]_2s_ Carbon-Fiber-Reinforced Polymer Laminates

**DOI:** 10.3390/mi14122209

**Published:** 2023-12-06

**Authors:** Yinghao Liao, Xin Liu, Changxi Zhao, Bing Wang, Liyan Zheng, Xiaoming Hao, Longxu Yao, Dian Wang

**Affiliations:** 1School of Power and Mechanical Engineering, Wuhan University, Wuhan 430072, China; yinghaoliao@whu.edu.cn; 2The Institute of Technological Sciences, Wuhan University, Wuhan 430072, China; xliu188@whu.edu.cn (X.L.); wang_dian@whu.edu.cn (D.W.); 3Beijing Spacecrafts, Beijing 100094, China; zhengliyan99@163.com (L.Z.); 13269215826@163.com (X.H.); 4School of Mechanical Engineering, Shandong University, Jinan 250061, China; sduyaolongxu@mail.sdu.edu.cn; 5Key Laboratory of High Efficiency and Clean Mechanical Manufacture of MOE, Key National Demonstration Center for Experimental Mechanical Engineering Education, Jinan 250061, China

**Keywords:** ultrasonic vibration, waterjet drilling, amplitude, delamination

## Abstract

The pure waterjet (WJ) drilling process of carbon-fiber-reinforced polymer (CFRP) laminates causes damage, such as tears and delamination, leading to poor-quality hole-wall. Ultrasonic-vibration-assisted technology can improve the quality of hole walls and repair such damage, particularly the delamination of CFRP laminates. In this study, we conducted a numerical and experimental investigation of a high-pressure pure WJ drilling process of CFRP laminates performed using ultrasonic vibration to improve the delamination phenomena of the pure WJ drilling process. An explicit dynamic model using the smoothed particle hydrodynamics method was employed to simulate the ultrasonic-vibration-assisted WJ drilling of CFRP laminates and ascertain the optimal drilling performance. Thereafter, WJ drilling experiments were conducted to verify the numerical simulation. The results illustrate that the employment of ultrasonic vibration significantly increased the material removal rate by approximately 20%. Moreover, the water-wedging action that induces the propagation of delamination was weakened with an increase in the amplitude of the ultrasonic vibration. The hole-wall quality was optimal with the following drilling parameters: amplitude, 10 μm; frequency, 20 kHz; and WJ velocity, 900 m/s. The delamination zone length was only 0.19 mm and was reduced by 85.6% compared with the values obtained using non-assisted WJ drilling.

## 1. Introduction

Carbon-fiber-reinforced polymer (CFRP) is an excellent material used in aerospace, automotive, and several structural applications [1], particularly in the machining of aircraft components. This is because of its remarkable mechanical and physical properties, such as low density, high specific strength, and high corrosion resistance [2]. Although adhesive bonding is frequently used to join CFRP laminate components [3], additional machining operations are often required to facilitate the further installation of rivets or bolted joints. For any mechanical fastening, conventional drilling (CD) has been extensively used to produce CFRP holes. However, compared with the metal materials used in the drilling process, various types of mechanical damage in terms of delamination, poor surface integrity, fiber pullout, fiber breakage, and burrs typically occur because of the heterogeneous and anisotropic nature of CFRP laminates during drilling [4,5].

Owing to the thermal or mechanical damage induced by the drilling process [6], waterjet (WJ) technology is regarded as an alternative to the CD process for CFRP laminate hole drilling [7]. WJ technology uses an ultrahigh-velocity WJ as a cutting tool [8]. The WJ impacts the target material and leads to the development of a stagnation region on the target material surface [9,10]. The water hammer effect caused by WJ particles makes a great contribution to material failure and removal [11]. Jeong and Jang [12] investigated the transient wave propagation of a WJ in a unidirectional and quasi-isotropic composite laminate and observed that the periphery of the jet spread faster than the energy release wave, thus, leading to the development of a stress wave in the workpiece material. Dunnen et al. [13] recorded the effect of nozzle diameter and bone architecture on hole dimensions. A mathematical prediction model has been established to describe the relationship between the hole depth and the nozzle diameter. Researchers have discovered that water-wedging action is the main agent that induces the delamination of CFRP laminates [14]. The anisotropic nature of CFRP laminates affects machining quality and leads to several uncontrollable damages, such as delamination, cracks in the matrix, and surface roughness, among which delamination is one of the most important defects [15]. To overcome the aforementioned problem, a less invasive drilling technique is required to mitigate damage during the WJ drilling of CFRP laminate components.

Ultrasonic vibration has been employed in tool machining [16] to improve the cutting performance for difficult-to-machine materials. During ultrasonic-vibration-assisted drilling, the periodic movement of the cutting tool with the target CFRP material, generated by a piezoelectric transducer, reduces the friction and cutting forces, thereby improving the drilling efficiency and the quality of the drilled holes. This technology has been extended to WJ drilling to improve processing efficiency and quality. Figure 1 shows a schematic of the ultrasonic-vibration-assisted WJ drilling process. The workpiece vibrates with the ultrasonic platform along the waterjet impingement direction. When the waterjet impinges on the surface of the workpiece, the high-frequency vibration induces the cavitation of the waterjet and increases the energy of the waterjet, thus contributing to the material removal of the workpiece. Qi et al. [17] employed computational fluid dynamics (CFD) numerical method to model the ultrasonic-vibration-assisted micro-channeling process on glasses using an abrasive slurry jet (ASJ). They observed that ultrasonic vibration was beneficial to viscous-flow-induced erosion, in turn enhancing the micro-channel top width and decreasing the channel wall inclination angle. Liu et al. [18] conducted a finite element method (FEM) analysis and observed that the vibration applied to a target workpiece surface adds no energy to the micro abrasive waterjet (AWJ) drilling process. Instead, the vibration rendered the transfer and application of the jet energy more effective. Lv et al. [19] adopted the smoothed particle hydrodynamics (SPH) method to build a SPH-FEM coupled model to investigate the erosion process of aluminum nitride material caused by ultrasonic-assisted AWJ machining. Their results show that ultrasonic vibration influences the dynamic process of particle penetration. When it comes to CFRP laminates, the mechanism of ultrasonic vibration in the WJ drilling process is still ambiguous. Additionally, the relationship between the ultrasonic vibration parameters and the initiation of delamination remains to be explored. It is also worth mentioning that the delamination factor is an important term used in composite industries to measure the quality of a hole [20]. In the CD process, the delamination factor (DF) and uncut fiber factor (UCFF) of the delamination area are introduced, serving as measures for the evaluation of peel up delamination at the hole entrance and push out delamination at the hole exit [15,20,21]. In this study, the initiation of delamination is caused by the water-wedging action, and the delamination is mainly extended along the inter-laminate interface; thus, the delamination zone length is introduced as the criterion for judging drilled hole quality instead of the DF and UCFF of the delamination area.

Here, the drilling process of CFRP laminates with a [0/45/−45/90]_2s_ stacking sequence subjected to an ultrahigh-velocity WJ with the assistance of ultrasonic vibration is investigated using a comprehensive SPH-FEM numerical model and a drilling experiment. The target CFRP material was modeled as a multilayer laminated composite to simulate the drilling process of CFRP laminates. Thereafter, a displacement function curve was introduced to simulate ultrasonic vibration on the target CFRP material. The vibration-assisted pure WJ drilling process of CFRP laminates is investigated numerically and experimentally to determine the mechanism of ultrasonic vibration in the WJ drilling process and the effect of ultrasonic vibration parameters on the initiation of delamination in CFRP and hole characteristics. The objective of this study is to enhance the comprehension of the ultrasonic-assisted WJ drilling process and determine the parameters required for achieving higher-quality holes.

## 2. Numerical Simulation

The WJ drilling process was modeled to study the hydrodynamic characteristics of a WJ under the effect of ultrasonic vibration. This study utilizes the commercial explicit dynamic solver LS-DYNA (LSTC, Livermore, CA, USA) to establish a three-dimensional (3D) coupled SPH-FEM model to model the process of an ultrahigh-velocity WJ impacting the target CFRP laminate under ultrasonic vibration. This was carried out to investigate the effects of ultrasonic parameters on hole quality and damages such as the delamination and cracking of CFRP laminates. Figure 2 shows geometric and finite element models of the numerical simulation.

The numerical model comprises a quarter of the WJ and target CFRP laminate material. The quarter model of the CFRP laminate comprises 16 plies with a stacking sequence of [0/45/−45/90]_2s_, resulting in a total thickness of 2 mm. The diameter of the WJ ejected from the nozzle was set to match the nozzle exit diameter, which is 1.0 mm (*R* = 0.5 mm). The surrounding boundary of the CFRP laminate material model was fixed using a clamp.

### 2.1. Modeling of Ultrasonic Vibration Waterjet Drilling

The WJ was modeled based on the mesh-free SPH method and expressed using numerous SPH particles. The NULL material model was introduced to describe the material properties of water. In this model, the thermal effect of water is ignored, and the Mie–Grüneisen equation of state (EOS) is employed to describe the behavior of water [22]. Table 1 shows the material and EOS parameters of water [23].

The effect of the air resistance acting on the WJ surface and the interaction of water particles and air are assumed to be negligible in this study, remaining at a standoff distance of 2 mm. Thus, the velocity of the jet impacting the target CFRP material surface equals the exit velocity at the nozzle.

The laminate composite continuum damage model based on a theory developed by Pinho et al. [24,25] was employed to predict the material removal process of the CFRP laminate with a mixed-ply stacking sequence. Table 2 lists the properties of the CFRP laminate material [26]. The cohesive zone model (CZM) has been widely used to simulate the delamination and fracturing of composite materials, particularly for laminate materials [27]. Here, the bilinear cohesive mixed zone method was introduced to simulate the interlaminate fracture behavior and delamination between the CFRP laminate plies [28]. Table 3 lists the properties of the cohesive interface for delamination damage [29].

Figure 1 also shows the geometric model and its boundary conditions. The symmetry boundary conditions were applied to the quarter model of the WJ and target laminate to economize computational resources. In addition, the symmetry boundary conditions were applied to the side walls perpendicular to the *X*- and *Y*-axes. The symmetry boundary constrains three degrees of freedom. For example, in the *Y*-*Z* plane, the translation in the *X*-axis and rotation along the *Z*- and *Y*-axes are restricted.

To investigate the effect of ultrasonic vibration on the characteristics of the WJ and the material removal process, dynamic motion was added to the target material model. The vibration direction is parallel to the velocity direction of the impacted jet. The motion of the ultrasonic horn can be described by kinematic differential equations. Under the conditions of harmonic vibration, the vibrating displacement of the target material is expressed as [30]
(1)$$z=A_{f}\sin(2\pi ft)$$
where *z* is the displacement of the target material moving along the *Z*-direction, *A_f_* is the vibration amplitude, and *f* is the vibration frequency. The ultrasonic vibration motion function was applied to the clamp model using the defined curve keyword based on Equation (1). The tied contact model was set with respect to the nodes between the clamp and material model to ensure that it was within the boundary conditions of the CFRP laminate material, thereby controlling the motion of the target material.

### 2.2. Mesh Independence Study

The effect of element mesh size on the maximum von Mises stress on the jet impingement on the workpiece surface was investigated to determine the optimal size of the mesh elements in terms of solution accuracy and computational time. The mesh of the WJ impingement core area was refined using hexahedral elements of five mesh sizes, and Figure 3 shows the results.

The element mesh sizes of 0.02 and 0.03 mm exhibit a discrepancy of 19.7% in predicting the von Mises stress; thus, the solution is dependent on the mesh size. Comparing the results of the mesh sizes of 0.01 and 0.02 mm, the maximum von Mises stress prediction differences are within 3%, demonstrating minimal solution dependency on the mesh. However, the computation time of the 0.01 and 0.02 mm meshes significantly differed such that the total computational times of the mesh sizes of 0.01 and 0.02 mm were 3652 and 1672 min, respectively. Thus, a mesh size of 0.02 mm was adopted.

## 3. Experimental Setup

In this work, we set up an experimental platform of ultrasonic-vibration-assisted pure waterjet drilling, as shown in Figure 4. CFRP laminates with a stacking sequence of [0/45/−45/90]_2s_ were drilled under ultrasonic vibration with various amplitudes and at various waterjet velocities.

The experiments were performed using the WL1520 waterjet machine produced by Wonlean Tech, Liaoning, China. The machine was equipped with a three-axis CNC cutting head with a positioning accuracy of approximately ±0.05 mm, while its cutting accuracy is approximately ±0.1 mm. The highest pressure produced using the extra-high voltage generator was 410 MPa. The ultrasonic vibration system comprised an ultrasonic generator, an ultrasonic transducer, an ultrasonic amplitude transformer, and an ultrasonic vibration platform. The ultrasonic vibration platform has dimensions of 200 × 200 mm. The output frequency of the ultrasonic generator is stable at about 20 kHz, and the output amplitude can be changed by adjusting the power of the ultrasonic generator. The ultrasonic vibration platform is equipped with an integrated amplitude transformer and an ultrasonic transducer, enabling the ultrasonic vibration platform to oscillate vertically in the Z-direction. Consequently, this allowed the CFRP laminate to undergo induced vibrations along the direction of jet movement with a specific amplitude.

The standoff distance (SOD) was set at 2 mm, aiming to minimize the influence of ambient air on waterjet flow. The effect of ultrasonic amplitude and waterjet pressure on WJ drilling performance were investigated and verified in a full factorial design of experiment (DoX). A jet impact angle of 90° and a nozzle diameter of 1.0 mm were kept constant during the experiment. A 2 mm thickness CFRP laminate with a [0/45/−45/90]_2s_ stacking sequence was employed.

## 4. Results and Discussion

The drilling processes of the CFRP laminate were simulated to determine the mechanism of ultrasonic-vibration-assisted WJ drilling, the effect of ultrasonic vibration on the WJ drilling performance, and the optimal processing parameters. The frequency of the ultrasonic vibration was set to 20 kHz during the simulation process. Various ultrasonic amplitudes (*A_f_*) (2 μm, 4 μm, 6 μm, 8 μm, and 10 μm) were considered during the simulation, and four velocity levels (600 m/s, 700 m/s, 800 m/s, and 900 m/s) of the WJ were used. Thereafter, a complementary experiment was conducted to validate the simulation results. A 3D laser confocal scanning microscope (VK-X200, KEYENCE, Osaka city, Japan) was used to monitor the delamination area, and the morphology of the hole surface was observed using a digital microscope (AD4113T, ANMO Electronics Corporation, Hsinchu City, China).

### 4.1. Process Characteristics of Ultrasonic-Vibration-Assisted Waterjet Drilling

Figure 5 shows the dynamic characteristics of the high-velocity pure WJ impacting the flat surface of the CFRP laminate. A relatively low WJ velocity in the *Z*-direction can be observed in the stagnation zone. This phenomenon can be attributed to the fact that when the WJ particles impinge on the flat surface of the material, it rebounds, and the back flow plays an important role in damping the velocity of the subsequent WJ. Thus, a stagnation zone is formed with a relatively low WJ velocity [17]. The contour of the velocity and particle trajectory demonstrate the aforementioned conclusion. As shown in Figure 3, the velocity of the material was set to 0, whereas the velocity of the incoming WJ was greater than the initial velocity (ignoring the effect of gravity). This is due to the high-frequency vibration with an amplitude of 10 μm on the CFRP laminate material. Thus, the WJ exhibited a higher incoming velocity, and the actual kinetic energy of the WJ increased.

Figure 6 shows that the numerical and experimental results of the distribution of delamination zones extended along the interlaminate interface of the CFRP laminate via the WJ impact at different initial velocities (*f* = 20 kHz and *A_f_* = 10 μm). It can be seen that with an increase in the WJ velocity, the machining time of the drilling process decreases from 35.5 to 13 μs. With an increase in the WJ velocity, the kinetic energy of the jet particle increases, resulting in a reduction in drilling time. Regarding the delamination area of the CFRP laminate drilled at varying WJ velocities, an increase in the WJ velocity causes a decrease in the delamination area of the CFRP laminate. The topography of the drilled hole sidewall observed using a 3D laser confocal scanning microscope shows the same tendency as that exhibited by the numerical results, and when the WJ velocity reached 900 m/s, the sidewall of the drilled hole barely exhibited a delamination area. The increase in the WJ velocity causes an increase in the kinetic energy of the WJ. When the WJ approaches the second or third layer, the high-speed WJ has surplus kinetic energy, and it is more likely that the elements of the second or third layer will be removed. Thus, the lateral flow is unlikely to induce the removal of the interlayer elements and the extension of delamination. Therefore, during ultrasonic-vibration-assisted hole drilling, a larger initial WJ velocity is beneficial for achieving high-quality holes.

Figure 7 compares the cross-sectional profiles of the holes drilled at various WJ velocities. The cross-sectional profile of the drilled hole can be approximated as a trapezoid. As the depth of the drilled holes increases, the hole width decreases. This phenomenon, known as the “kerf angle”, has been obtained in the experimental results of other studies [31]. It is characteristic of the WJ drilling process and demonstrates the accuracy of the simulation method. Delamination occurs in the longitudinal and transverse directions, as determined from the comparison of the *X*-*Z* and *Y*-*Z* planes, and the damage is asymmetrical owing to the anisotropic nature of CFRP laminates. When the WJ impinges on the CFRP laminate, the surface erodes, and the material is removed. The cohesive elements are also removed during this process, thereby inducing the initiation and expansion of the delamination area. The WJ wedges into the interlaminate interface and contributes to the warping of the CFRP laminate, particularly under the impingement of the WJ at velocities of 700 and 800 m/s. The delamination area decreases and the quality of the cross-sectional profile improves with an increase in the WJ velocity.

### 4.2. Effect of Ultrasonic Vibration on Waterjet Drilling Performance

Figure 8 shows the effects of ultrasonic vibration on the velocity of the WJ flow at an initial WJ velocity of 900 m/s. When the high-velocity WJ impinged on the CFRP laminate, not all the WJ energy contributed to the removal of the material. When the WJ penetrated the CFRP laminate material, part of the WJ flow rebounded off of the target, and the divergent flow may have contributed to the back flow and lateral flow [32]. Once delamination had been initiated on the side wall, the lateral flow may have wedged into the crack and enlarged the delamination (Figure 8). The velocity components of the WJ in the *X*- and *Y*-directions without ultrasonic vibration are near 500 m/s, inducing a larger lateral flow and an enlarged delamination area. Furthermore, as shown in the von Mises stress distribution, the stress of elements near the lateral flow exceeded that of other elements. Contrarily, the velocity components of the WJ in the *X*- and *Y*-directions with ultrasonic vibration were considerably lower, and the delamination area was reduced. Thus, high-frequency ultrasonic vibration plays a role in decreasing lateral flow, reducing the delamination area.

Figure 9 shows the effects of ultrasonic vibration on the MRR and hole quality of the CFRP laminate. The machining times of the drilling processes without and with ultrasonic vibration are approximately 16 and 13 μs, respectively, with an MRR increase of approximately 18.7%. This is because the energy of the ultrasonic vibration causes an increase in the kinetic energy and impact velocity of the high-speed WJ [33]. Therefore, the CFRP laminate elements are rapidly removed, and the MRR increases. On the other hand, without ultrasonic vibration, a larger amount of energy is transferred to the lateral flow, which wedges into the interlaminate interface and enlarges the delamination. Thus, ultrasonic vibration can provide energy to the WJ and help to reduce the energy loss of divergent flow, such as lateral flow, which improves the MRR.

Figure 9 also compares the quality of the sidewalls of the drilled holes. Without ultrasonic vibration, the lateral flow wedges into the interlaminate interface and reduces the quality of the hole sidewall. With ultrasonic vibration, the quality of the drilled hole sidewall is better, and hydro-wedging action hardly occurs, thereby reducing the generation of delamination zones. The effect of amplitude changes on hole quality was investigated. Figure 10 shows the distribution of delamination zones extended along the interlaminate interfaces of the CFRP laminate under ultrasonic vibration at various amplitudes. With an increase in the ultrasonic vibration amplitude, the extension of the delamination zones and the number of interlaminate interfaces exhibiting delamination decrease. When the amplitude of the ultrasonic vibration was increased from 2 to 10 μm, the energy of the ultrasonic vibration increased. Therefore, the velocity and kinetic energy of the WJ indirectly increased, improving the quality of the drilled-hole sidewalls.

Figure 11 shows the surface morphologies of the drilled-hole sidewalls observed using a confocal scanning microscope. Without ultrasonic vibration, a severe delamination area appeared, and the quality of the drilled hole sidewalls became poor, especially at the bottom orifice. With the increase in the vibration amplitude, the delamination area decreases, which is consistent with the phenomenon presented in the numerical results. It can also be seen in Figure 11 that the sidewalls of the drilled hole may be subjected to excessive material removal, leading to the poor quality of the sidewall. This can be attributed to the lateral flow (as shown in Figure 8). Some other micro-defects, such as fiber pull-out, fiber fragment, and bending failure, can also be observed in Figure 11.

Figure 12 shows a morphological comparison of the erosion surface with and without ultrasonic assisted drilling. The roundness error of the top orifice drilled with ultrasonic vibration assistance is a mere 0.036 mm, whereas without vibration assistance, the roundness error increases to 0.08 mm. Similarly, the roundness of the bottom orifice drilled with ultrasonic assistance exhibits superior results. Additionally, the distribution of cracks on the erosion surface corresponding to ultrasonic-assisted hole drilling is noticeably reduced. The cracks on the erosion surface display greater length and higher density in the absence of ultrasonic assistance. This phenomenon can be attributed to the lower stress distribution on both the top and bottom surfaces of the CFRP laminate. The application of ultrasonic vibrations can significantly enhance erosion surface quality.

Figure 13 shows the cross-sectional profiles of holes drilled at various ultrasonic vibration amplitudes at a machining time of *t* = 16.0 μs. The cross-sectional profile along the *X*–*Z* plane significantly differs from the profile along the *Y*–*Z* plane. The cross-sectional profile along the *Y*–*Z* plane can be approximated as a trapezoid, particularly the cross-sectional profile of the drilled hole associated with ultrasonic vibration at an amplitude of 4 μm along the *X*–*Z* plane. The back flow and lateral flow of the WJ contributed to the removal of the inner-layer elements outside the impact zone, resulting in poor-quality drilled holes. For the six profiles of holes drilled at varying amplitudes, the delamination area decreases with an increase in the amplitude. At an ultrasonic vibration amplitude of 10 μm, the profile of the WJ-drilled CFRP laminate hardly exhibits a delamination zone. Thus, ultrasonic vibration can significantly improve hole quality, and optimal hole quality can be obtained at an ultrasonic vibration amplitude of 10 μm.

To further assess the quality of the drilled holes, the maximum delamination zone length was extracted from the simulation result data. Figure 14 shows the changes in the delamination length and machining time with the WJ velocities and ultrasonic vibration amplitudes. As shown in Figure 14a, the maximum delamination zone length decreases from 3.82 to 0.19 mm with an increase in the WJ velocity, demonstrating that the increase in the WJ velocity is beneficial to the drilling process of the CFRP laminate. With an increase in the WJ velocity, the drilling time also decreases from 35.5 to 13.0 μs. The WJ velocity can significantly affect efficiency because an increasing WJ velocity implies a significant increase in the energy input in the processing system. However, regarding the maximum delamination zone length of CFRP laminates, the section of the WJ velocity from 700 to 800 m/s levels off because of a balance between the energy used to generate the delamination zone and the energy required for material removal in the depth direction. The maximum delamination zone length of the CFRP laminates significantly decreases with a further increase in speed. As shown in Figure 14b, with an increase in the ultrasonic vibration amplitude, the maximum delamination zone length of the CFRP laminates decreases from 1.36 to 0.19 mm at a WJ velocity of 900 m/s. The drilling time slightly decreases with an increase in the ultrasonic vibration amplitude. The larger the vibration amplitude, the greater the WJ’s energy. Thus, the WJ has a stronger removal effect in the depth direction. This reduces the maximum delamination zone length of the CFRP laminates, which ultimately reduces the energy loss during delamination to improve the MRR. The results demonstrate that an amplitude of 10 μm and a WJ velocity of 900 m/s afford optimal MRR and hole quality.

## 5. Conclusions

In this study, the WJ drilling process of CFRP laminates aided by ultrasonic vibration was numerically and experimentally investigated. A numerical model was developed by using the coupled SPH–FEM method to simulate the ultrasonic-vibration-assisted WJ drilling of CFRP laminates and determine the optimal drilling performance. The user-defined displacement function was introduced to simulate high-frequency ultrasonic vibration on the target CFRP laminate. In addition, the CZM was employed to simulate the effect of the interlaminate interface on the initiation and propagation of the delamination zone. Thereafter, WJ drilling experiments were conducted to verify the numerical simulation. The following conclusions could be drawn:Ultrasonic vibration contributes to the energy of the WJ penetration of the CFRP laminate. During the WJ drilling of the CFRP laminates, ultrasonic vibration significantly increases the MRR by approximately 20%.Ultrasonic vibration can significantly improve the damage zones and decrease the delamination zone length with an increase in amplitude. The water-wedging action inducing the propagation of delamination is weakened with an increase in the amplitude of the ultrasonic vibration due to the decrease in the lateral flow.The quality of the hole wall is optimal at an amplitude (*A_f_*) of 10 μm, a frequency (*f*) of 20 kHz, and a WJ velocity of 900 m/s. The delamination zone length is only 0.19 mm and was reduced by 85.6% compared with the WJ drilling without ultrasonic vibration.

This study facilitates an understanding of ultrasonic-vibration-assisted material removal mechanisms and effects on the hole wall quality, thereby guiding the high-quality WJ drilling of CFRP laminates.

## Figures and Tables

**Figure 1 micromachines-14-02209-f001:**
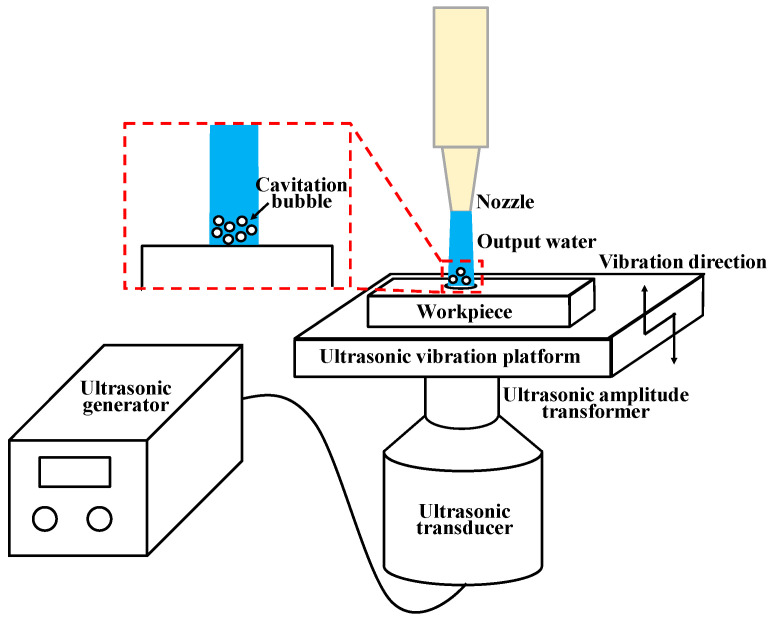
Schematic of ultrasonic-vibration-assisted WJ drilling process and mechanism of ultrasonic vibration.

**Figure 2 micromachines-14-02209-f002:**
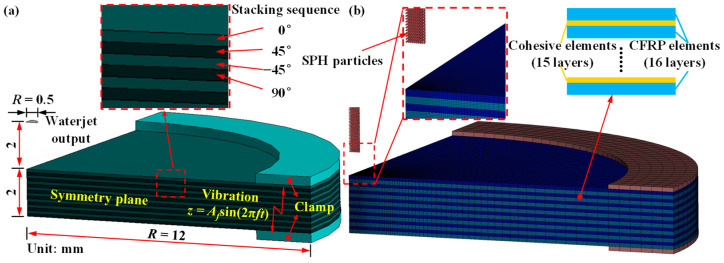
(**a**) Geometric and (**b**) finite element models of ultrasonic-vibration-assisted waterjet (WJ) drilling.

**Figure 3 micromachines-14-02209-f003:**
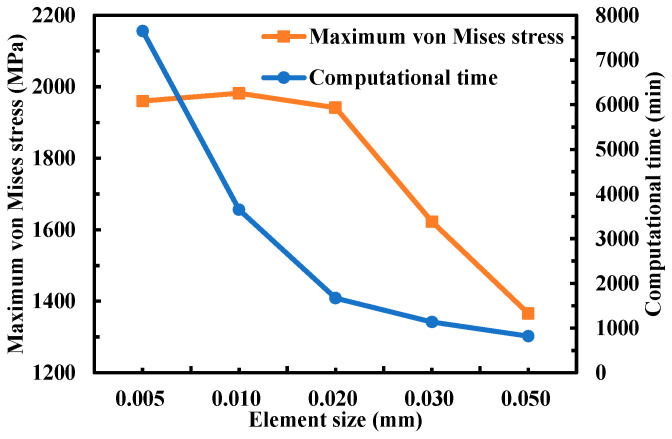
Mesh independence study.

**Figure 4 micromachines-14-02209-f004:**
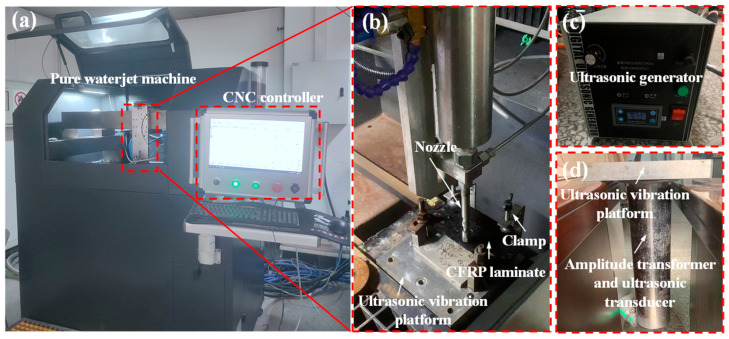
Ultrasonic-vibration-assisted waterjet system. (**a**) Pure waterjet machining center. (**b**) Cutting head and processing platform. (**c**) Ultrasonic generator. (**d**) Ultrasonic-vibration-integrated platform.

**Figure 5 micromachines-14-02209-f005:**
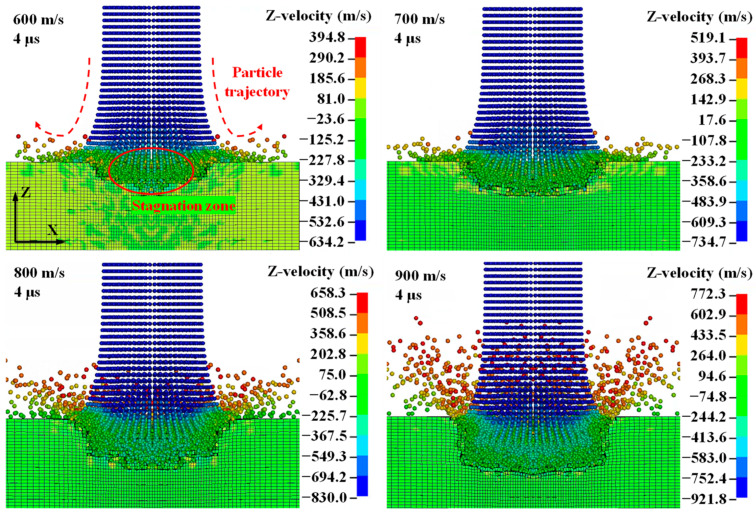
High-velocity pure WJ impacting the flat surface of the CFRP laminate with ultrasonic vibration (vibration amplitude, 10 μm; frequency, 20 kHz).

**Figure 6 micromachines-14-02209-f006:**
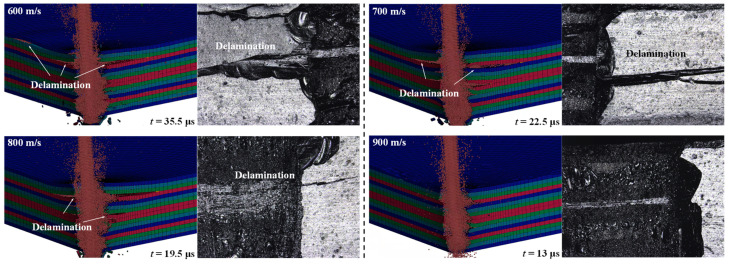
Distribution of delamination zones along the interlaminate interface of the CFRP laminate by the impact of the WJ at various initial velocities (amplitude, 10 μm; frequency, 20 kHz).

**Figure 7 micromachines-14-02209-f007:**
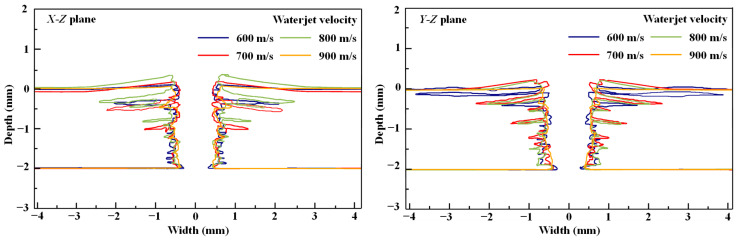
Cross-sectional profiles of the drilled holes at varying WJ velocities.

**Figure 8 micromachines-14-02209-f008:**
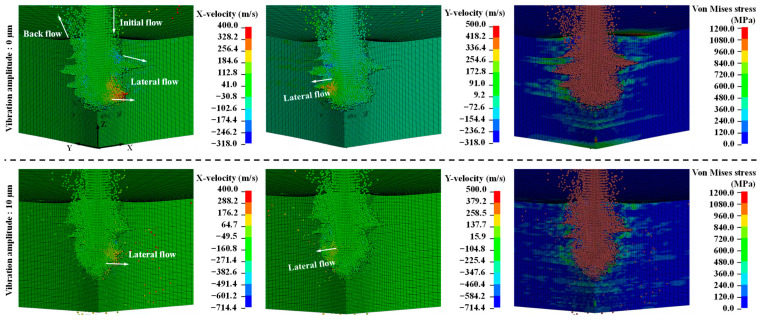
Effects of ultrasonic vibration on WJ velocity (initial waterjet velocity, 900 m/s; frequency, 20 kHz).

**Figure 9 micromachines-14-02209-f009:**
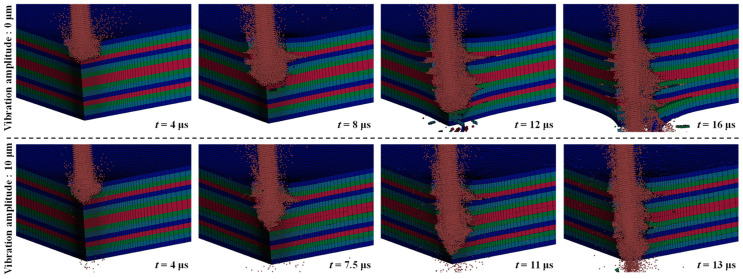
Effects of ultrasonic vibration on the MRR and hole quality.

**Figure 10 micromachines-14-02209-f010:**
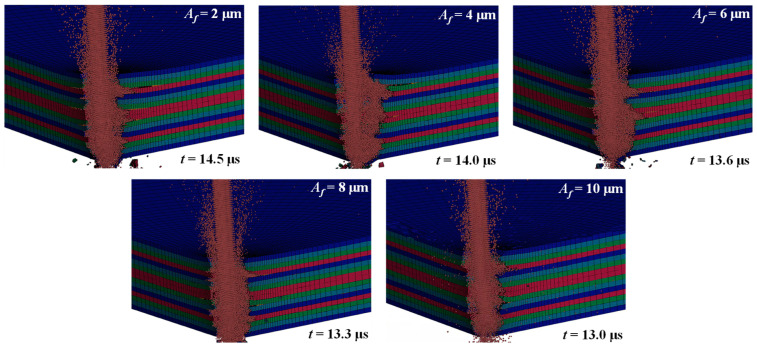
Distribution of delamination zones extended along the interlaminate interfaces of the CFRP laminate under ultrasonic vibration at various amplitudes (waterjet velocity, 900 m/s; frequency, 20 kHz).

**Figure 11 micromachines-14-02209-f011:**
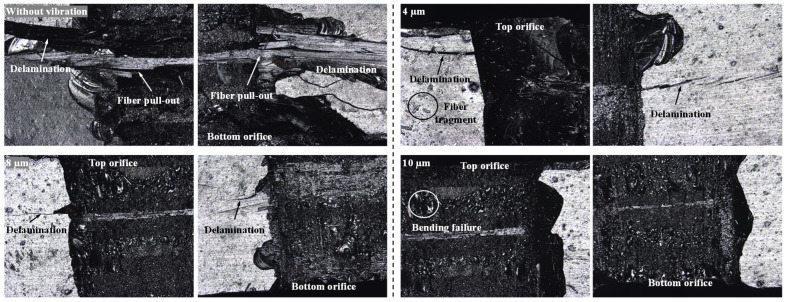
Surface morphologies of the drilled-hole sidewalls varied with the ultrasonic vibration amplitudes (WJ velocity, 900 m/s).

**Figure 12 micromachines-14-02209-f012:**
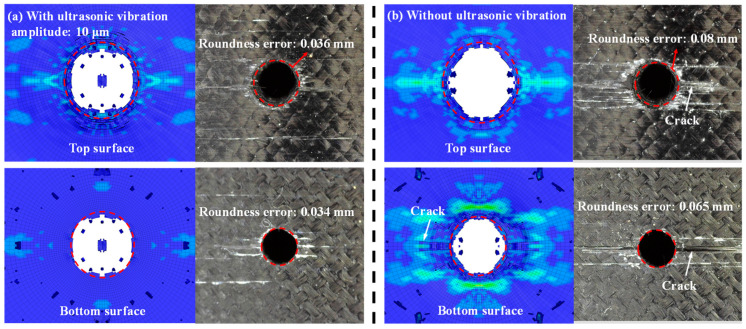
The morphological comparison of the erosion surface (**a**) with and (**b**) without ultrasonic assisted drilling (WJ velocity, 900 m/s).

**Figure 13 micromachines-14-02209-f013:**
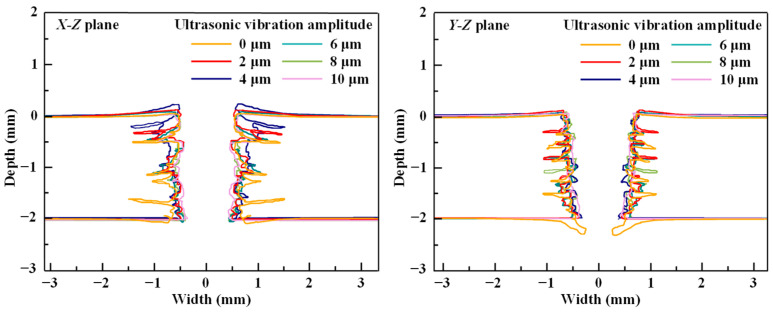
Cross-sectional profiles of holes drilled at varying ultrasonic vibration amplitudes (Waterjet velocity, 900 m/s; frequency, 20 kHz; impingement time, 16 μs).

**Figure 14 micromachines-14-02209-f014:**
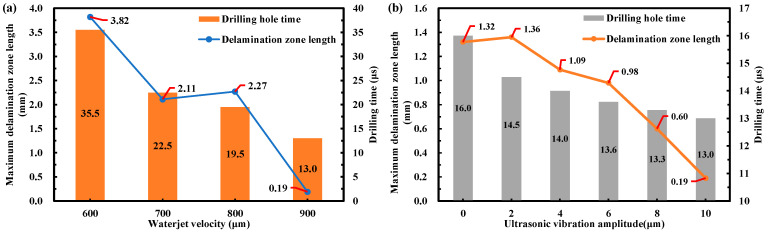
Changes in delamination length and machining time with the WJ velocity and ultrasonic vibration amplitude (**a**) with ultrasonic vibration at a constant frequency of 20 kHz and an amplitude of 10 μm and (**b**) a drilling WJ velocity of 900 m/s and ultrasonic vibration at a constant frequency of 20 kHz.

**Table 1 micromachines-14-02209-t001:** Material and EOS parameters of water [23].

Description	Parameters
Mass density	1000 kg/m^3^
Dynamic viscosity	1.02 × 10^−3^ Pa·s
Cut-off pressure	−1.0 × 10^20^ Pa
Speed of sound in water, C	1483 m/s
Grüneisen gamma, *γ*_0_	0.28
Coefficient, *S*_1_	1.75

**Table 2 micromachines-14-02209-t002:** Material parameters of the CFRP laminate [26].

Parameter	Value
Longitudinal Young’s modulus, *E_a_*	127 GPa
Transverse Young’s modulus, *E_b_*	9.40 GPa
Out-of-plane Young’s modulus, *E_c_*	9.40 GPa
Poisson’s ratio, *ν_ba_*	0.019
Poisson’s ratio, *ν_ca_*	0.019
Poisson’s ratio, *ν_cb_*	0.4
Density	1.49 g/cm^3^
Shear modulus, *G_ab_*	4700 MPa
Shear modulus, *G_bc_*	3100 MPa
Shear modulus, *G_ca_*	4700 MPa
Longitudinal compressive strength, *X_C_*	1082 MPa
Longitudinal tensile strength, *X_T_*	2231 MPa
Transverse compressive strength, *Y_C_*	100 MPa
Transverse tensile strength, *Y_T_*	29 MPa
Shear strength, *S_C_*	60 MPa

**Table 3 micromachines-14-02209-t003:** CFRP material properties for delamination modeling [29].

Parameter	Value
Maximum traction in mode I, *T*	44.8 MPa
Maximum traction in mode II, *S*	60.4 MPa
Interface stiffness, *EN*, *ET*	10^5^ N/mm^3^
Mode I fracture toughness, *G_IC_*	0.105 kJ/m^2^
Mode II fracture toughness, *G_IIC_*	0.5 kJ/m^2^

## Data Availability

The datasets used or analyzed during the current study are available from the corresponding author on reasonable request.

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
