# Peer review of "Ultrasonic-Vibration-Assisted Waterjet Drilling of [0/45/−45/90]2s Carbon-Fiber-Reinforced Polymer Laminates"

_micromachines, 2023, doi:10.3390/mi14122209_

Round 1

Reviewer 1 Report

Comments and Suggestions for Authors

my comments on the article are attached

Reviewer 2 Report

Comments and Suggestions for Authors

It is an original paper studying numerical research to explore the use of ultrasonic vibration in a high-pressure pure waterjet (WJ) drilling process for CFRP laminates. The goal was to enhance the drilling process by reducing delamination issues associated with pure WJ drilling.

·       The novelty of the work must be clarified in the last paragraph of the introduction. What is the gap in this research area?

·       The introduction needs to be revised. Add a schematic image to explain the process and critical components. How does the ultrasonic help the waterjet drilling process?

·       When you are saying the use of ultrasonic vibration to improve the drilling quality, what do you mean? Is there any quality control criteria to monitor the drilled hole quality? For example, for quality control of drilled holes in composites, there is some research that introduces criteria to make a comparison between two drilled hole at the top and bottom surfaces of composites. I recommend you discuss this and see if it is possible to discuss this in your work. Here are some of the research introducing these criteria, “Optimization of drilling parameters in composite sandwich structures (PVC core)”, “Influence of machining parameters on delamination in drilling of GFRP-armour steel sandwich composites”, and “Defect evaluation of the honeycomb structures formed during the drilling process”

·       Table 2 needs to be referred to properly.

·       Equation 1 needs to be referred to properly.

·       In this research, the effect of ultrasonic amplitude and velocity was investigated. What about stacking sequence, distance of nozzle to the surface, nozzle diameter, etc.?

·       Use bullets to highlight the main achievements of the work in the conclusion section.

·       How accurate is this model? Have you done any experiments to verify your simulation?

Reviewer 3 Report

Comments and Suggestions for Authors

The article presents a numerical investigation focusing on the high-pressure pure Water Jet (WJ) drilling process applied to Carbon Fiber Reinforced Polymer (CFRP) laminates. The study utilizes ultrasonic vibration as a novel technique aimed at mitigating delamination issues associated with the conventional WJ drilling process. The primary objective is to enhance the understanding of ultrasonic vibration's role in improving the removal of CFRP laminates, and subsequently, to optimize drilling performance. The chosen research topic is highly relevant, given the increasing importance of CFRP materials in various industries. The integration of ultrasonic vibration to enhance the drilling process presents an innovative approach with significant potential applications. However, to ensure the article's suitability for publication, several key recommendations should be considered:

1.      In the introductory section, the authors have provided a comprehensive review of the literature on vibration-assisted waterjet drilling with ultrasound. Nevertheless, they have failed to explicitly state their unique contribution in terms of the literature cited. It is important that they clearly state the distinctiveness of their research compared to existing studies and clarify why their work stands out from others in the field.

2.      Authors should check the font of the letters in the tables, that is, they should format them according to the instructions for writing the paper. Table 2 should be mentioned earlier in the text.

3.      It is advisable that the authors include a visual representation of the ultrasonic vibration-assisted waterjet drilling system in Section 2. This should explain which elements within the system are subject to vibration and the specific parameters associated with the ultrasonic vibrations. Details such as frequency, amplitude, and other relevant information.

4.      Lines 182 and 184 begin with the same phrases. The authors should change that.

5.      The main drawback of the manuscript is that it is based only on theoretical research. Which does not give it a strong scientific contribution. Although it has already been said that the subject is quite topical, especially from the point of view of the application of CFRP in industry, a purely theoretical approach without confirming experiments does not make a significant contribution to the field.

6.      The paper in its current state may not meet the journal's publication standards. The authors should include experimental results to support their claims and increase credibility.

Round 2

Reviewer 2 Report

Comments and Suggestions for Authors

The authors have tried to respond to the comments properly. There are still some points that need to be addressed before publication. 

You have responded that "In order to further quantify the drilled hole quality, we have counted the maximum delamination zone length". The delamination length is not the best criterion for investigating the damage that occurred in a laminate (the damaged area is more practical). Therefore, as commented earlier, discuss the below publications in the introduction section. And explain why delamination length was selected instead of delamination area.

·       “Optimization of drilling parameters in composite sandwich structures (PVC core)”,

·       “Influence of machining parameters on delamination in drilling of GFRP-armour steel sandwich composites”,

·       “Defect evaluation of the honeycomb structures formed during the drilling process”

Reviewer 3 Report

Comments and Suggestions for Authors

After a thorough review and analysis, I found that the authors have adeptly addressed all of my suggestions and comments. Consequently, I firmly believe that the manuscript is now poised for publication.

Author Response

Thanks again for your profound suggestions and comments